# Amyloids: The History of Toxicity and Functionality

**DOI:** 10.3390/biology10050394

**Published:** 2021-05-01

**Authors:** Elmira I. Yakupova, Liya G. Bobyleva, Sergey A. Shumeyko, Ivan M. Vikhlyantsev, Alexander G. Bobylev

**Affiliations:** 1Institute of Theoretical and Experimental Biophysics, Russian Academy of Sciences, Pushchino, 142290 Moscow, Russia; liamar@rambler.ru (L.G.B.); shumik92@gmail.com (S.A.S.); ivanvikhlyantsev@gmail.com (I.M.V.); bobylev1982@gmail.com (A.G.B.); 2A. N. Belozersky Institute of Physico-Chemical Biology, Lomonosov Moscow State University, 119991 Moscow, Russia

**Keywords:** protein aggregation, amyloids, functional amyloids, amyloidosis, amyloidogenesis, Alzheimer’s disease

## Abstract

**Simple Summary:**

This review presents the beginning of the history of toxic properties of amyloids, especially on Aβ amyloids. We discuss anti-amyloid therapy and its problems and write about new views on amyloids that can play positive roles in different organisms including human.

**Abstract:**

Proteins can perform their specific function due to their molecular structure. Partial or complete unfolding of the polypeptide chain may lead to the misfolding and aggregation of proteins in turn, resulting in the formation of different structures such as amyloid aggregates. Amyloids are rigid protein aggregates with the cross-β structure, resistant to most solvents and proteases. Because of their resistance to proteolysis, amyloid aggregates formed in the organism accumulate in tissues, promoting the development of various diseases called amyloidosis, for instance Alzheimer’s diseases (AD). According to the main hypothesis, it is considered that the cause of AD is the formation and accumulation of amyloid plaques of Aβ. That is why Aβ-amyloid is the most studied representative of amyloids. Therefore, in this review, special attention is paid to the history of Aβ-amyloid toxicity. We note the main problems with anti-amyloid therapy and write about new views on amyloids that can play positive roles in the different organisms including humans.

## 1. Introduction

The history of the study of amyloidosis dates back to the 17th century, when a woman was found to have a greatly enlarged spleen, which was crudely cut out with a knife [1]. In later works of the 19th century, the term “wax liver” was used for an organ in which large quantities of an unknown substance had collected [2]. This is the point from which researchers started studying amyloids deposited in different organs of the body, affecting the liver, spleen, kidneys and so forth [2,3]. Such diseases have subsequently been named “amyloidosis”.

Amyloids are defined as aggregates of misfolded peptides or proteins, with a quaternary cross-β structure, due to which they possess properties such as insolubility and resistance to proteolysis [4]. Because of these properties, accumulation/deposition of amyloids may occur in the body, thereby resulting in considerable changes in the metabolism of tissues and organs [4]. Amyloids are believed to lead to the development of amyloidoses. Amyloidoses are a group of incurable diseases. More than 30 different amyloidogenic proteins are currently associated with various diseases in humans [1,4]. Some of these are well known, for example AD. It is not only amyloids that are found in specific deposits in tissues with amyloidosis. Proteins such as glycosaminoglycans, apolipoprotein E and serum amyloid P components are also often present [4].

Due to the research carried out by Prusiner and his colleagues in the 1980s, the nature of infectious diseases such as Creutzfeldt–Jakob disease in humans, scrapie in sheep and bovine spongiform encephalopathy became better known [5,6]. It turns out that these diseases are caused by proteins called prions (or rather their altered form). Prions are a unique class of infectious agents. All of these diseases are related to the prion protein (PrP), which can change its conformation from a non-infected form (PrPSc) to an infected one [7]. This form is similar to an amyloid cross-β conformation. Therefore, we can say that another class of amyloid diseases (prion diseases) has emerged.

Since the beginning of the 21st century, our knowledge of amyloids has advanced, thanks to data on protein aggregates with an amyloid structure that do not cause dis-ease and may even have a specific role in the body. These are therefore called functional amyloids. Functional amyloids have been found in plants [8,9], prokaryotes [10,11,12] and eukaryotes [13,14,15,16,17,18,19], including mammals [16,17,18] and even humans [17,18]. All groups of organisms have adaptations that protect them from the toxic effects of amyloids. Here, we present a new perspective on amyloids that can play a positive role in different organisms.

## 2. Toxic Properties of Amyloids (the Beginning of Their History)

In the literature, amyloids are presented as being the main cause of development of diseases such as AD, amyloidosis of the liver and kidneys, etc. There is the main “cascade amyloid hypothesis” formulated by Hardy and Higgins (1992) [20]. According to this hypothesis, the cause of AD is formation and accumulation of amyloid plaques from fragments of the Aβ amyloid precursor protein (APP), which cause a cascade of molecular events, leading to loss of cell function and their death [20]. It was initially shown that Down syndrome (trisomy 21) patients have an extra copy of the amyloid precursor gene on chromosome 21. Their Aβ levels are also high, and, by age 40, they are likely to exhibit clinical symptoms of AD [21,22]. The second discovery was the familial type of AD, caused by mutations in the APP that lead to overproduction of Aβ [23,24,25,26,27]. Aβ is a 4.2-kDa short peptide of 40–42 amino acids, which is generated from intracellular cleavage of the APP by the sequential action of β-secretase and γ-secretase proteolytic enzymes [28].

In AD, two types of aggregates are found: extracellular Aβ deposits and intracellular tau protein. The toxicity of intracellular aggregates seems to be due to the sequestration of crucial proteins, together with amyloid, which leads to loss of cellular function and even cell death [29]. In AD, intracellular tau aggregates can, in theory, trap functional proteins and tau itself, which might induce microtubule destabilization [29]. Hardy’s hypothesis implies that, in AD, accumulation of Aβ-amyloids is the primary cause of neuronal death, and, for a post-mortem diagnosis, it is necessary to detect deposits of aggregated Aβ [20]. This is why Aβ-amyloid is the most studied representative of amyloids. Therefore, in this review, special attention is paid to Aβ-amyloid.

Aβ toxicity has been demonstrated on cultured nerve cells in vitro [29,30,31,32]. A study in 1989 showed that a peptide derived from the APP was toxic for hippocampal neurons in a culture [29]. Another study reported that a peptide ligand homologous to the first 28 residues of the Aβ-amyloid protein (Aβ1–28) increased the survival of cultured hippocampal neurons and appeared to have a neurotrophic effect [30]. In 1990, Whitson et al. detected that synthetic peptide β1–42 has a neurite-promoting effect, including extensive dendritic branching and axonal elongation [33]. In addition to the neurotrophic effect, a neurotoxic effect of Aβ has also been observed in other experiments [31]. For instance, in 1990, Yankner et al. showed that a portion of the amyloid beta protein, consisting of APP amino acids 25–35, contributed to trophic and, at the same time, toxic effects [31]. 

An article published in 1991 reported that the discovered in vitro effects could be caused by protein self-aggregation [32]. Data on the spontaneous assembly of a full-length β-amyloid peptide (β1–42) have already been reported in existing studies [33,34,35]. Based on this evidence, Pike et al. were perhaps the first researchers to assume that self-aggregation of β1–42 determines whether peptide trophic or toxic properties will be found [32]. To verify the hypothesis, Pike et al. demonstrated that β1–42 incubation influences the viability of hippocampal cultures in vitro. They found that pre-incubated β1–42 caused a neurotoxic effect, while recently solubilized β1–42 caused no toxicity but did promote neurite outgrowth [32]. Furthermore, toxicity of aggregates of the peptide that was incubated at 37 °C for 24 h increased the cell death effect [32,33]. Strong neurotoxicity in certain peptide sequences in Aβ1–42 (for example, β25–35), which exhibited rapid aggregation, was detected immediately upon solubilization [30]. Thus, it was concluded that Aβ toxicity is dependent on concentration, with low doses being protective and high doses being toxic [36]. A low dose of Aβ may also protect cells from the effects of a subsequent high dose [36].

The next stage in the history of Aβ toxicity included the hypothesis that negative effects on neurons are dependent on the ratio of Aβ1–42 to Aβ1–40 peptides. Aβ42:Aβ40 ratios of 3:7 (1 and 2 h) and 10:0 (1 and 2 h) were found to be associated with neurotoxicity in cultured mouse hippocampal neurons in vitro and to influence memory formation in mice in vivo. Aβ42:Aβ40 ratios of 1:9 and 0:10 did not show neurotoxicity at any stage of the aggregation process [37].

In many of the above studies, the toxic effect of Aβ was observed. However, what is the mechanism of βAP-mediated neurotoxicity? It is known that many aspects of calcium homeostasis change with aging (1987) [38] and that imbalances in calcium regulation lead to neural degeneration (1992) [39]. On this basis, in a 1993 study, it was suggested that one of the mechanisms behind the toxic effect of Aβ is disturbance of Ca^2+^ homeostasis [40]. Mattson et al. [40] noted that an experimental increase in [Ca^2+^] can induce antigenic and ultrastructural changes in neurons similar to those observed in neurofibrillary tangles in AD [32,41].

Continuing the research, Mattson concluded (in 1994) that Aβ which arises from alternative processing of beta APP aggregation destabilizes Ca^2+^ homeostasis [42]. Furthermore, vulnerable neurons possess high levels of glutamate receptors [42], and several growth factors can stabilize Ca^2+^ and protect neurons against excitotoxic injury and Aβ toxicity [42]. In the same year (1994), Weiss et al., studying the neurotoxicity of Aβ in primary murine cortical neurons, reported that Aβ (fragment 25–35) led to neurodegeneration that is concentration-dependent, and this effect decreased in the presence of Ca^2+^ channel blockers such as nimodipine (1–20 mM) and Co^2+^ (100 mM) [43]. In this context, neuronal Ca^2+^ homeostasis dysregulation is considered to be a common factor that underlies AD pathogenesis [44]. This effect is possible due to several mechanisms: direct interaction with membranes and further destabilization of the membrane structure [44,45,46,47,48]; formation of a cation-conducting pore [49,50,51,52,53,54,55]; and activation of cell surface receptors coupled with Ca^2+^ influx [56,57,58]. More recent studies also support the hypothesis of toxicity of Aβ amyloids, associated with changes in calcium homeostasis [59,60,61,62,63,64,65]. It is also assumed that neurotoxicity of Aβ might occur via generation of reactive free radicals [65,66,67,68,69,70,71,72]. In 1992, the cytotoxic effects of Aβ and an internal fragment encompassing residues 25–35 were shown using cultured cortical nerve cells. Vitamin E (known as an antioxidant and free radical scavenger) was found to inhibit Aβ-induced cell death [68]. At the same time (1992), five brain tissue sections from an AD case and five normal age-matched controls were studied with polyclonal antibodies against superoxide dismutase (CuZn- and Mn-forms) and catalase immunostaining [72]. As a result, a subgroup of neurofibrillary tangles (15–25%) and senile plaques (50%) showed immunoreactivity for both enzymes [72,73]. In 1994, Behl et al. suggested that Aβ causes increased levels of H_2_O_2_ [74,75]. 

In 1994, research provided evidence that beta-amyloid can inactivate oxidation-sensitive glutamine synthetase and creatine kinase enzymes [67]. A 1995 study investigated the release of nitric oxide (NO) from cultured rat microglia exposed to synthetic Aβ25–35 and Aβ1–40 (alone or in combination with cytokines IFN-α/β, IL-1β, TNF-α or TNF-β, including IFN-γ) [70]. Only IFN-γ among the cytokines being tested induced release of NO from the cells. β25–35 did not stimulate the release of nitric oxide by itself, but, in combination with IFN-γ, it caused NO release. In fact, β1–40 triggered NO production in microglia by itself, and this effect increased in conjunction with IFN-γ (100 U/mL) [70]. In 2020, it was shown that docosahexaenoic acid can suppress oligomeric Aβ-induced reactive oxygen species [71].

Further evidence indicating that generalized oxidative stress is important in pathogenesis effects was provided by a study of hippocampal cells. The study demonstrated that tyrosine nitration is increased in neuronal cytoplasm as well as in the nuclei of both neurons and glia in the regions of AD pathology. Thus, there is strong evidence that peroxynitrite is involved in oxidative damage of AD pathology [68,69].

In 1997, using primary cultures of cerebellar granule cells and astrocytes, it was shown that nanomolar concentrations of peptides such as Aβ-(1–40) and Aβ-(25–35) potently activate transcription factor NF-κB [76]. The authors of this work also conducted an immunohistochemical analysis of brain slices from patients with AD. In this case, activation of NF-κB was also observed in neurons and astroglia. Activated NF-κB was observed near the plaques [76]. NF-κB activation can lead to neuroprotection [37] via inhibition of amyloid β-mediated neuronal apoptosis [37].

Apoptosis is known to play a role in many neurologic disorders, including AD [77,78,79]. Dickson noted some problems in earlier experiments on apoptosis in AD: (1) the data were observed on cell cultures in vitro, which cannot necessarily be extrapolated to organisms in vivo; (2) in cell culture research, very high concentrations of Aβ are usually used, which do not exist in nature [80]; and (3) evidence for frank cellular apoptosis in AD is controversial [81].

Aβ is the main candidate for activation of apoptotic mechanisms in AD. It has recently been shown that insoluble Aβ complexes, including protofibrils and oligomers, play a role in this process [82]. Findings indicate that Aβ can activate caspases [83,84,85,86,87,88,89,90,91,92]. This mechanism can be triggered by the accumulation of Aβ at the site of its synthesis in endoplasmic reticulum or endosomes, which causes activation of apoptotic mechanisms through the unfolded protein response or endoplasmic reticulum stress. Apoptosis may also occur if Aβ activates alcohol dehydrogenase through mitochondrial stress [92].

The hypothesis of the toxic effects of APP protein fragments, having first undergone changes in favor of the toxic effects of Aβ aggregates (fibrils), has now been revised in favor of the highest toxicity of oligomers. It has already been demonstrated that Aβ oligomers exhibit neurotoxicity and cause neuronal damage [93,94,95,96,97,98,99,100,101,102,103,104,105,106,107]. Aβ protofibrils prepared in in vitro conditions have a smaller diameter than amyloid fibrils and high β-sheet content. It has been shown that a genetic risk factor for AD (APOJ; encoding clusterin) promotes formation of oligomeric structures with soluble properties [94]. Such oligomeric assemblies are involved in cell death both in vitro and in vivo [95,96]. A study in which soluble Aβ oligomeric species were extracted from the brain tissue of patients with AD showed that the presence of soluble oligomeric species correlates more closely with symptoms of the disease than with fibrils from amyloid plaques [97,98].

Mutations of the gene encoding APP are known to cause the development of AD [99]. It has been shown that two clinical mutations in the APP gene can increase the tendency of Aβ to oligomerize [100,101]. Benilova et al. concluded that, at the time of their study (2012), further evidence was needed to support the oligomeric hypothesis [93]. Indeed, there are a number of difficulties associated with research into oligomeric forms of the peptides. Firstly, the solution contains a mixture of water-soluble Aβ species [93]; and, secondly, as shown by Bitan et al., sodium dodecyl sulfate (SDS) can actually artificially induce oligomerization of Aβ [102]. It is known that the SDS-PAGE method used for obtained monomers, trimers and tetramers (as major bands) [102] is not a reliable method. The presence of SDS therefore makes it difficult to extrapolate in vivo effects from in vitro evidence [93]. It is still necessary to show whether similar modifications of Aβ monomers can occur in vivo. Some modifications of the peptide that can occur in vivo under the action of certain enzymes have already been discussed in the literature [104,105,106], but whether neurotoxicity in AD is associated with such modifications has not been fully elucidated.

The mechanisms behind the toxic effects of Aβ oligomers on cells may also lead to disruption of calcium homeostasis, and this has yet to be discussed in the literature [107]. The effects of oligomers of Aβ from both extracellular and intracellular Ca^2+^ sources have been shown [95,107]. The researchers also noted that Aβ42 and its oligomers caused increased membrane permeability in general [107]. This can cause unregulated flux of ions and molecules and, in turn, may be a common mechanism of oligomer toxicity [107]. However, we should note that high oligomer concentrations were used for the experiment to increase Ca^2+^ signals (100–1000 times greater than the levels of soluble Aβ measured in the brain of AD patients [107]).

Oligomers can show toxicity through membrane, intracellular and receptor-mediated mechanisms [78,108]. A study has been conducted on Aβ synaptotoxic effects, including disturbances in the functioning of N-methyl d-aspartate (NMDA) receptors that can affect calcium influx [79,108]. However, whether Aβ oligomers have a direct influence on NMDA receptors is a matter of controversy [97]. Alternative pathways suggest that Aβ induces synaptic failure due to apoptotic pathway activation [109] or upregulation of nicotinic acetylcholine receptor α7-nAcChR51 [109]. There is evidence of interactions with the insulin receptor [110]. Furthermore, the effects on the hypoxia-induced factor, clustering of angiotensin type 2 receptor and other mechanisms have also been described [111,112]. Some studies have also measured induction of apoptosis or other markers of cell death [108]. It is also interesting that astrocyte cell cultures have been found to be resistant to the action of Aβ [109]. Several reports in particular show that Aβ deposits in the brain do not necessarily correlate with AD symptoms [113,114]. Thus, we can conclude that there is still no consensus on the main mechanism of the toxic effects of amyloid fibrils or/and Aβ oligomers.

## 3. Anti-Amyloid Therapy and Its Problems

Amyloidogenesis, amyloid aggregates found in AD and the disease itself have gained increasing attention in the search for therapeutic drugs. Therapeutic agents/molecules have various properties, including the ability to do the following [111]:(1)Block β-sheet formation(2)Prevent fibrillogenesis(3)Dissolve Aβ aggregates into non-toxic species(4)Destabilize Aβ oligomers(5)Accelerate the conversion of Aβ oligomers to Aβ aggregates (modulators of Aβ aggregation)

However, despite the fact that amyloids are considered to be the main cause of AD development, therapeutic agents designed to reduce Aβ-peptide production or prevent Aβ-peptide aggregation have been found to be ineffective in phase III clinical trials [110,115]. At the same time, the number of developments in the search for drugs to treat AD has already exceeded 2300 [115]. However, AD remains an incurable disease.

Recent comparative studies of amyloid aggregates formed in vitro and isolated from tissues indicate morphological differences in amyloids. In particular, brain-derived amyloid fibrils of Aβ-peptides are right-twisted, while in vitro analogs are left-twisted [112]. This indicates that the study of amyloidogenesis in in vitro models has its own limitations, and the properties of “in vitro aggregates” may differ from their “in vivo analogs”. Moreover, the study of amyloidogenesis in vivo is a very complex task, requiring development of new research methods. All this raises the critical question of whether it is correct to extrapolate results obtained from in vitro studies relating to amyloids to in vivo systems. At the same time, the hypothesis about the main role of amyloids in development of AD is not confirmed and does not come without criticism [111,116,117].

The above data once again cast doubt on the fact that amyloids are harmful to the body and do not play any positive role in cells. In this regard, it is necessary to highlight interesting data showing that amyloids in brain cells accumulate at a certain time of the day in all healthy people (picomolar concentrations) [118,119]. This is believed to have some positive effect [111]. In favor of this assumption, we include reference to the results of experiments demonstrating that, when drugs capable of reducing the formation of amyloids are used, neurons die. In one study, neuronal death was prevented by adding amyloids [120]. It has been shown that all known developed drugs with an anti-amyloid effect cause death in people [121,122]. For example, active Aβ1–42 immunization (with AN-1792) resulted in 6% of patients developing meningoencephalitis [123]. At the same time, a reduction in senile plaques was observed [124], and none of these patients showed improved cognitive functioning [20]. Thus, should amyloids be treated as enemies or friends, and is it necessary to get rid of them?

To answer these questions, one of the strategies of anti-amyloid therapy, namely blocking the formation of β-sheets, should be discussed. This approach, in turn, depends on two membrane enzymes: β-secretase, also known as the β-site of the APP-cleaving enzyme (BACE1), and γ-secretase [23], consisting of five subunits, namely Pen-2, Presinilin 1 or Presinilin 2, Nicastrin and Aph1B. γ-secretase is involved in catalyzing the formation of several peptides, for instance, the Notch1 signal peptide, which plays a role in growth and proliferation of cellular processes [24]. It has been shown that subunit Presenilin 1 is important for the production of Aβ-peptide. One experiment found that knockout of the γ-secretase gene (in subunit Presenilin 1) caused the death of mice already at the embryonic stage [25]. Furthermore, inhibition of the entire γ-secretase gene by non-selective inhibitors led to various pathologies associated with Notch1 formation, including gastrointestinal tract disorders, impaired functioning of the immune system and skin pathologies [25,26]. In earlier studies, enzyme complex modulators instead of γ-secretase inhibitors were used to cause a shift in the production of Aβ-peptides from toxic Aβ fragments (1–42) to less toxic and shorter ones [20]. Inhibition of β-secretase with knockout of the gene in mice led to interruption of the accumulation of Aβ-peptides [21,27,28]. Mice lacking β-secretase secretion reproduced relatively healthy offspring with minor phenotypic abnormalities, such as hypomyelination [29,30] and behavior with schizophrenia-like symptoms [30]. Some experiments have focused on the search for selective inhibitors of BACE1. It has been shown that inhibitor OM99-2 polypeptide is capable of penetrating the blood–brain barrier, and, in 2008, clinical studies of this polypeptide began [31]. This inhibitor has a limitation, i.e., a “bulky” peptide structure for in vivo conditions [31]. A drug was subsequently created on the basis of OM99-2, which was able to stay in the bloodstream for a longer time, pass through the blood–brain barrier and reduce the level of Aβ-peptides (30–65%) in the plasma of transgenic mice modeling AD [32,39]. More than 10 drugs have been developed with similar characteristics. One of these has been found to be particularly successful (CTS-21166) and was tested on a group of volunteers suffering from Alzheimer’s disease. These drugs are promising, but none of them have gone through all the stages of clinical research. Most of the tested compounds were withdrawn from the initial stages of clinical studies for various reasons. Thus, are amyloid deposits the main cause of AD, and do their Aβ toxic properties really cause the death of nerve cells? 

Understanding the exact mechanism of toxicity caused by Aβ oligomers or aggregates/fibrils (or both) is still a matter of discussion. In 2012, Benilova et al. presented an interesting point of view, asserting that proof of this “invisible toxin” is comparable to the (in)famous teapot of the British philosopher Bertrand Russell [28]. Results from clinical tests on anti-amyloid drugs also suggest that the approach adopted in AD treatment needs to be reviewed. To do this, the multifactorial nature of this disease needs to be taken into account. While researchers pay full attention to amyloid deposits, they ignore the mechanisms of development of cytoskeleton abnormalities, inflammation, oxidative stress, other metabolic abnormalities, etc. (Figure 1) [21,125,126].

We should not forget that the brain is extremely sensitive to fluctuations in blood glucose concentrations, and a lack of glucose for more than a few minutes can ultimately lead to cell death [127]. It is known that hypoglycemia [128], inadequate transportation of glucose across the blood–brain barrier [129], defective astroglial glutamate transport [130] and hyperammonemia [131] are associated with mental function. Comparison of the brains of patients who aged normally with the brains of AD patients has revealed certain metabolic differences, including glucose metabolism and cellular ATP, which decrease in the case of AD patients [132]. This supports the significant role of cerebral energy metabolism in AD pathology [119,133,134,135]. Thus, the cause of AD cannot only be due to amyloids but also other factors.

There is a hypothesis about the possible protective role of Aβ production in AD [136]. In the case of head trauma, neuroinflammation, ischemia-hypoxia/reperfusion, anesthesia or infectious agents, which are characteristically associated with neuronal damage [137,138], Aβ may exhibit trophic and neuroprotective [30], antioxidant [30,138] and antimicrobial properties [139]. Aβ can also be observed in tissue after metabolic impairment by non-specific stimuli [135]. 

Glucose and oxygen deregulation have been observed in patients with AD [126,140,141,142]. It has even been suggested that reduced mental capacity is caused by an imbalance of metabolic processes rather than the effects of Aβ deposits [135]. Red blood cells in the brains of AD patients have poorer oxygen transport efficiency than those in patients with normal brains [143]. A study conducted by Kosenko et al. on metabolites and enzymes in the main metabolic pathways [135] reported an increase in red blood cell glycolysis and ion fluxes in AD [137,144].

With AD, reduced brain glucose availability results in an increase in the cerebral ammonia concentration [145,146]. Moreover, ammonia is a powerful neurotoxin, and its accumulation in AD brains might be the reason for deterioration of memory and cognitive abilities.

It should also be noted that there is chronic inflammation in the brains of AD patients, and several components are involved. Activation of the brain’s resident macrophages (microglia) and specific cytokine signaling occurs in AD [147].

In 1992, immunohistochemical studies showed co-localization of the C1q component of the complement system, with amyloid deposits, in the brain of AD patients [148]. Researchers concluded that Aβ amyloids can directly activate the complement system. After this, in vitro studies started to research direct activation by amyloid fibrils in the absence of antibodies, according to the classical pathway [148,149,150,151,152,153,154,155]. Such activation was also observed for the alternative pathway [156,157,158]. It has been demonstrated that complement activation by Aβ fibrils in vitro results in the formation of C5a, as well as assembly of the C5b-9 proinflammatory membrane attack complex [152,155]. It has been shown that many complement system proteins, including C1q, C4, C3, C5, C6, C7, C8 and C5b-9, are also located near Aβ deposits and neurofibrillary tangles in the brains of AD patients [150,152,158,159,160,161,162]. This was followed by application of obtained data to new therapeutic approaches for reducing neuroinflammatory damage by influencing of the complement system response [161]. Recent in vitro studies do not confirm a direct antibody-independent complement system activated by Aβ fibrils [162]. However, the finding that complement system proteins are co-localized with amyloid deposit plaques needs to be explained further. It is worth noting that in addition to Aβ peptides, amyloid plaques contain other components, such as glycosaminoglycans, apopolyprotein E and amyloid serum component P [4]. Nevertheless, in this study [4], an association with the complement system was found only in the case of Aβ peptides. This raises the question of which factor it activates. Increased bacterial populations in the brain of patients with AD, compared with a normal brain, have been observed [163,164]. Therefore, Firmicutes, Actinobacteria and, in particular, P. acnes can directly play a role in neuroinflammation. In addition to co-localization of amyloids with the complement system, it has also been shown that amyloid aggregates have an antimicrobial function [164]. Thus, the complement system is more likely to be activated by microorganisms than by Aβ fibrils or other components of amyloid inclusions [162]. However, microbiota can influence the development of neurodegenerative diseases through amyloid and lipopolysaccharide formation in the gut, which trigger an increased inflammation response in the brain [165,166].

Thus, we can see that the primarily negative attention given to amyloids (especially Aβ) is changing now. There is no one uniform toxicity mechanism for Aβ and other amyloids. We still do not know why or how proteins/peptides aggregate in vivo and the reason for this process. In this review, we pay close attention to the history of Aβ toxicity in order to explain how researchers have studied amyloids and have come to a dead end, especially in AD treatment. Amyloids have still only been the subject of in vitro investigations. There are animal models of AD, but they are still far removed from human pathology. We highlight the main problems with anti-amyloid therapy above and next consider a new perspective on amyloids that can play a positive role in the functioning of different organisms, including humans.

## 4. Useful Properties of Amyloids. Functional Amyloids

Evidence that amyloids may have positive properties has been reported, and we now know that amyloids have antioxidant [137,138] and antibacterial effects [4,139,164]. Many amyloids are produced in the body, and it has been discovered that they play a vital role. Such amyloids are called functional amyloids. The first description of amyloidosis probably appeared in 1639 [1]. The first functional amyloid was discovered at the turn of the 20th century [10,167,168,169] (Figure 2). To date, more than 20 functional amyloids have been described. 

Functional amyloids have various roles (Table 1), from formation of dense hydrophobic monolayers on the surface of spores and fruiting bodies of some fungi [166] to formation of the carcass of spider silk [15]; from protection of melanin toxic intermediates in melanosomes [16,17] to formation of long-term memory in animals [16,170]. They are involved in the fertilization process in mice [17] and regulation of the synthesis (or content) of hormones in humans [19]. Recent data indicate that amyloid aggregates can participate in immune responses because they are part of the extracellular neutrophil trap [171]. Functional amyloids are found in muscle tissue. It has been shown that, during regeneration of skeletal muscles in mice and humans, the cytoplasmic RNA-binding protein TDP-43 forms amyloid-like oligomers called myogranules [172]. Amyloids have also an RNA-modulating ability and play a role in transcription, translation, storage and degradation of RNA [173]. It has been shown that some motifs in prions of fungi and humans can be functionally related and be a model of amyloid signaling mechanisms from fungi to mammals [174,175]. Prokaryotic cells can also use such properties [8]. Just as people can store hormones in amyloids, plants can also store their protein using amyloid formation [8].

What is the reason for the non-toxicity properties of functional amyloids? Why can one be pathological but another functional? To date, the mechanisms behind the non-toxicity properties of functional amyloids are still unclear. However, Jackson and Hewitt suggested realistic ways to resolve this: (1) regulating the content of amyloidogenic peptides/proteins; (2) decreasing the time of the oligomers state during amyloidogenesis; (3) locating amyloids within membrane-bound organelles (e.g., melanosoma); and (4) regulating amyloid formation by other molecules and disassembling the fibrils under physiological conditions [229].

Might it be better to use the aggregation properties of amyloids? Scientists have now discovered how amyloid aggregates can inhibit VEGFR2-dependent tumor growth in a mouse tumor model [213], which was also observed using different human cell lines [226]. Oligomers can influence tumor growth too [230]. It has been shown that amyloids have antimicrobial properties [163,231], and these properties can be realized to β-sheet ion channel formation by amyloids [232].

## 5. Conclusions

The discovery of amyloids in organs and tissues in various diseases has tarnished the reputation of these formations for many years. Until now, amyloids have been considered to be harmful pathogenic aggregates, affecting cell homeostasis and, ultimately, resulting in cell death. There are numerous data on the toxicity of amyloids (Aβ amyloids in particular). Meanwhile, recent studies report that amyloids may have a beneficial role in cells and in organisms, and not just contribute to the development of amyloidosis, including AD. The results obtained in these experiments will enable many researchers to refocus their efforts on diligent study of the structural features and functional properties of amyloids. This will also yield important data to support selection of the most appropriate approaches for treatment of amyloidosis.

## Figures and Tables

**Figure 1 biology-10-00394-f001:**
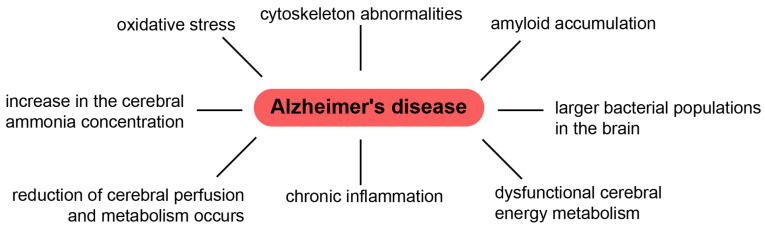
The mechanisms of AD.

**Figure 2 biology-10-00394-f002:**
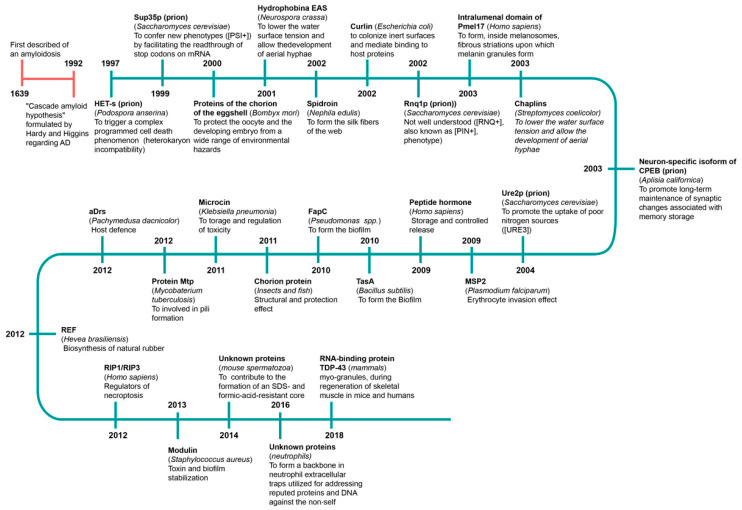
Timeline of history of functional amyloid discoveries. The name, host and function are presented.

**Table 1 biology-10-00394-t001:** The prevalence of functional amyloids and features of their research [176,177,178,179,180,181,182,183,184,185,186,187,188,189,190,191,192,193,194,195,196,197,198,199,200,201,202,203,204,205,206,207,208,209,210,211,212,213,214,215,216,217,218,219,220,221,222,223,224,225,226,227,228].

Species or Organisms	Protein or Peptide	Function	Mol. Weight	Structure	Evidence of Cross β-Structure Presence	Secondary Structure Changes	Congo Red and ThT Binding	Condition of in vitro Amyloid Fibril Forming	References
**Bacteria**									
*Escherichia coli, Salmonella spp.*	Curli	Biofilm formation, host invasion.	CsgA (main damain of curlin) ~17.5 kDa.	Previously β-structure.	X-ray diffraction for CsgA.	CD method: CsgA fibrils are as follows: 16 ± 2% α-helix, 40 ± 2% β-sheet, 13 ± 2% β-turn and 31 ± 2% remainder.	CR, ThT	CsgA fibrils were prepared by dialyzing purified protein into 25 mM Tris, pH 7.5, 100 mM NaCl and 0.5 mM EDTA and incubating at room temperature (RT) for several days.	[176,193,194,195]
*Streptomyces coelicolor*	Chaplins	Modulation of water surface tension (i.e., development of aerial structures).	ChpD-H up to 6 kDa ChpA-C ~17–20 kDa.	ChpD and ChpF comprise β-sheet; ChpE is random coil (RC); ChpG and ChpH have mixed secondary structure comprising elements of both β-sheet and RC.	X-ray diffraction.	CD method: the protein mixture adopted a conformation rich in β-sheet.	ThT	Synthetic chaplin peptides were dissolved at a final concentration of 0.5 mg/mL in water and the pH adjusted by titration of NaOH/HCl.	[11,196]
*Rhizobium leguminosarum*	RopA and RopB	Possibility role in the control of plant-microbial symbiosis.	RopA 38.97 kDa RopB 22 kDa.	Previously β- structure.	none	CD method: Before aggregation: RopA more than 40% β-structure, RopB more than 30% β-structure After aggregation: 42% and 38% β-structure for RopA and RopB aggregates respectively.	CR, ThT	Proteins were dissolved in 1,1,1,3,3,3-hexafluoro-2-propanol (HFIP) and incubated for seven days. Afterward, HFIP was evaporated under a stream of nitrogen, and the samples were stirred for an additional seven days.	[197]
*Klebsiella pneumoniae*	Microcin E492 (Mcc)	Bacteriocin, membrane pore-forming peptide, amyloid form is inactive.	~7.8 kDa.	RC conformation in aqueous buffer and α-helix in methanol.	X-Ray diffraction.	CD method: Aggregated Mcc is rich in β-sheet structures.	CR, ThT	Purified Mcc a (400 μg/mL) were incubated in aggregation buffer (50 mM PIPES-NaOH, pH 6.5, 0.5 M NaCl) for 48 h at 37 °C with vigorous shaking.	[198,199,200]
Xanthomonas species	Harpins (HpaG)	Secreted by plant pathogenic bacteria, destabilize plant membranes, induce cell death.	15.6 kDa.	Previously α-helix.	non	CD method: After 3 days, the CD spectrum of HpaG changed to a minimum at 220 nm, which is indicative of transition to a β-sheet.	CR	Harpin samples were incubated without agitation in 20 mm Tris-HCl (pH 8.0) containing 10 mm NaCl to mimic the salt concentration in the intercellular space of plant tissues at 27 °C for 14 days.	[201]
**Fungi**									
*Podospora anserine*	HET-s	Regulation of heterokaryon formation.	~32 kDa.	Estimated content of 34% α-helical, 16% β-sheet and 50% RC structure.	X-ray diffraction of HET-s (218–289).	CD method: 17% α-helix, 32% β-sheet and 50% random coil. FTIR: In the infrared spectrum of the soluble form the amide I′ band reached a maximum at 1650 cm^−1^. In the spectrum of the aggregated form this maximum was shifted at 1643 cm^−1^ and a shoulder around 1625 cm^−1^ was observed.	CR, ThT of HET-s (218–289).	The HET-s (218–289) peptide was soluble at pH 2.5 in 150 mM acetic acid, but, under non-denaturing conditions at pH 8.0, in a time course of a few hours, the peptide spontaneously formed aggregates.	[202,203,204,205]
*Saccharomyces cerevisiae*	URE2p	Regulation of nitrogen catabolism.	~38 kDa.	β-strands, α-helix and RC.	Electron diffraction, X-ray diffraction and X-ray diffraction (PFD domain).	CD method of PFD domain: Switching from an initially disordered, RC structure, to a β-sheet enriched conformation. FTIR of PFD domain: A band at ∼1625 cm^−1^ dominates the spectrum (the presence of intermolecular β-sheet structure).	CR, ThT	Filaments were made by incubation of protein solutions (usually at about 1 mg/mL) on a shaker for 16 h at 4 °C.	[206,207]
	Sup35p (Prion-inducing domain 2–114 and PFD domain)	Regulation of stop-codon read-through.	~75 kDa.	A freshly prepared solution exhibits a far UV CD spectrum that indicates little α-helix or β-sheet content.	X-ray diffraction (PFD domain).	CD method (PFD domain): Switching from an initially disordered, RC structure, to a β-sheet enriched conformation. FTIR: A band at ∼1625 cm^−1^ dominates the spectrum.	CR	Filaments of Sup35pN (Prion-inducing domain 2–114) were prepared in 0.1% (vol/vol) TFA/40% (vol/vol) acetonitrile using reverse-phase HPLC fractions containing isocratically eluted Sup35pN. Preparation of a 100 μM solution of Sup35pN yielded filaments after 1 week of incubation at 4 °C. Spontaneous filament formation exists in 50 mM sodium phosphate buffer (pH 2.0) with 40% acetonitrile.	[207,208,209]
	Swi1p	Chromatin remodeling factor, prion form inactive.	~140 kDa.	none	X-ray diffraction (PFD domain).	CD method (PFD domain): Switching from an initially disordered, RC structure, to a β-sheet enriched conformation. FTIR: A band at ∼1625 cm^−1^ dominates the spectrum.	none	none	[209,210]
	Mot3	Transcriptional regulator of cell wall remodeling genes, prion form is inactive.	~55 kDa.	none	X-ray diffraction (PFD domain).	CD method (PFD domain) Switching from an initially disordered, random coil structure, to a β-sheet enriched conformation. FTIR (PFD domain) A band at ∼1625 cm^−1^ dominates the spectrum.	CR, ThT.	none	[207,211]
Most fungi	Hydrophobins	Fungal coat formation, modulation of adhesion and surface tension.	7–9 kDa.	Previously RC and small core of antiparallel β-sheet.	X-ray.	CD method: β sheet is the predominant element of secondary structure in polymerized hydrophobin rodlets.	CR, ThT	For Hydrophobin SC3 Schizophyllum commune: Upon binding to a hydrophobic solid surface, the protein is arrested in an intermediate α-helical state, whereas, upon self-assembly at the air–water interface, rodlets are formed in a β-sheet conformation.	[176,212,213,214]
**Animal**									
Insects and fish	Chorion proteins (central domain of silkmoth chorion proteins of the A and B -family)	Structural and protective functions in the eggshell.	34 and 24 kDa.	In both families of proteins β-sheet structure predominates.	X-ray diffraction	FTIR method ATR FT-IR supports the presence of uniform β-sheets in the structure of cA_m1 peptide fibrils; β-sheet structure also suggested by X-ray diffraction ATR FT-IR data: 64% antiparallel β-sheet and 30% β-turns in the central domain of silkmoth chorion proteins.	CR	cA peptide (central domain of the A class of silkmoth chorion proteins) was dissolved in a 50 mM sodium acetate buffer (pH 5) at a concentration of 9 mg/mL to produce amyloid-like fibrils after 3–4 weeks incubation.	[14,215,216,217,218]
*Nephila clavipes* *Nephila edulis* *Araneus* *diadematus*	Spidroins and Araneus diadematus fibroin	Structural (i.e., spider silk).	~320 kDa (spidroin).	β-sheet or β-turn and RC.	X-Ray diffraction.	CD method: increasing of β-sheet structures.	CR, ThT	Lyophilized protein was dissolved in 6 M guanidinium thiocyanate at a concentration of 10 mg/mL^−1^ and dialyzed against 10 × 10^−3^ M potassium phosphate for several days at RT. For acceleration of fibril formation, 10 vol.-% methanol was added.	[15,177,219]
All mammalians including *Homo sapiens*	Non-glycosylated, 442-residue lumenal fragment of Pmel17 (rMα)	Pmel17 amyloid templates and accelerates the covalent polymerization of reactive small molecules into melanin.	110 kDa (28-kDa transmembrane fragment (Mβ) and an 80-kDa lumenal fragment (Mα)).	β-strands, α-helix and RC	X-ray diffraction	CD and FTIR: Mα aggregates are approximately 11% α-helix, 32% β-sheet, 23% β-turn and 33% disordered, based on curve fitting with a basis set of 43 soluble proteins.	CR, ThT	rMα fibers were generated by diluting (from concentrated 8 M GdmCl, 50 mM KH2PO4/K2HPO 4 [pH 7.4], 100 mM KCl stock) rMα into 125 mM CH3COOH/ CH3COOK buffer (pH 5.0) at a final concentration of 10 μM and allowing it to stand at RT for 24 h.	[18,220]
*Drosophila melanogaster*	CPEEB (Orb2)	Memory consolidation Cytoplasmic polyadenylation element-binding protein regulates mRNA translation.	~62 kDa	The protofilament core adopts a simple hairpin-like fold, composed of two β-strands, b1 (residues 176 to 186) and b2 (residues 197 to 206).	CryoEM	Only 31 residues (176–206) of the 704-residue protein form the amyloid core. N650 residues are dynamically disordered.	ThT	Recombinant Orb2A and Orb2A88 samples were exchanged into 10 mM HEPES, pH 7.6, 100 mM KCl, 1 M Urea and 1 mM DTT using dialysis and a PD-10 desalting column, respectively. Samples were then incubated on a shaker at RT for up to 2 weeks.	[221,222,223,224]
**Plants**									
*Pisum sativum* L.	Vicilin (Cupin-1.1 ((19–166 aa) and Cupin-1.2 (229–394 aa))	Amyloid formation in charge of the accumulation of storage proteins in plant seeds.	~50 kDa.	β-barrel domains.	X-Ray diffraction.	CD method Before aggregation Cupin-1.1 and Cupin-1.2 (4–12% β-content), Vicilin (39% β-content) After aggregation Cupin-1.1,Cupin-1.2 and Vicilin (40–42% β-content).	CR, ThT.	1,1,1,3,3,3-Hexafluoro-2-propanol (HFIP) solvent for the proteins dissolution with its subsequent removal from the sample and incubation of dissolved proteins in the distilled water at 37 °C for 7 days for Vicilin, Cupin-1.1, Cupin-1.2 and 5 mM phosphate buffered saline (PBS) [pH 7.4]) for one day at 25 °C) for Cupin-1.2	[8]
**Synthetic amyloid aggregates**									
Synthesized peptides (*Homo sapiens* and *Mouse)*	Vascin (Peptide based on an amyloidogenic sequence in the vascular endothelial growth factor receptor (VEGFR2)	Inhibited VEGFR2-dependent tumor growth.	2272.15 Da.	Secondary structures are absent.	X-Ray diffraction.	FTIR method b-sheet structure change in b-structured conformation.	ThT.	300 mM vascin in 1% (w/v) NH_4_CO_3_ after 24 h incubation at room temperature.	[225]
*Dosidicus gigas (D. gigas)*	Sucker ring teeth (SRT) from squid. SRT are assembled entirely from a protein family	Molecular design of biomimetic protein- and peptide-based thermoplastic structural biopolymers with potential biomedical and 3D printing applications.	none	Previously β-structure.	X-Ray diffraction.	FTIR method: the β-sheet-specific infrared band was centered at 1.235 cm^−1^ from RT up to 150 °C, at which point it shifted gradually to 1.220 cm^−1^, which was still within the β-sheet- region.	none	none	[226]
Synthesized peptides	Gonadotropin-releasing hormone analog (GnRH)	Use of amyloids in the formulation of long-acting drugs. Sorting, storage, and release of diverse hormones.	1183.27 Da.	Secondary structures are absent.	none	none	CR, ThT.	GnRH analogs were dissolved in a glass tube in 1 mL of 5% D-mannitol and 0.01% sodium azide at a concentration of 1 mg/mL. The GnRH analogs were then incubated at RT without stirring.	[227]
*Escherichia coli* and *Bacillus circus* and *Mytilus galloprovincialis*	CsgA (as amyloidogenic cores) + chitin-binding domains (CBDs) + mussel foot proteins (Mfp3/Mfp5) two-domain and three-domain constructions with constant presence of CsgA	Development of multifunctional molecular materials with individual structure and characteristics based on amyloid.	CsgA ~17.5 kDa Mfp3 5–7.5 kDa Mfp5 9.5 CBD 6 kDa.	β-strands and RC.	X-Ray diffraction.	The two-domain proteins contained 60% of β-sheet/β-turn structures and 40% of RC, owing to the introduction of RC Mfps. Compared with their two-domain counterparts, the three-domain fibrils possess more β-sheet structures.	CR, ThT	Proteins were either dialyzed against PBS solutions (pH = 5.0 or 2.5) for 2 days or were incubated at 4 °C under acidic conditions for 3 days to promote the formation of amyloid fibers, followed by redissolving in hexafluoro-2-propanol (HFIP) solvent.	[228]

## Data Availability

Not applicable.

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
