# Peer review of "Amyloids: The History of Toxicity and Functionality"

_biology, 2021, doi:10.3390/biology10050394_

Round 1

Reviewer 1 Report

The authors sought to provide a detailed view of Ab in AD. I felt that a lot of effort was contributed to achieve this goal, and this manuscript is clear logic and well-written. However, it is mandatory to explain/correct the manuscript in some points.  

  1. Several important papers should be included in the introduction to provide sufficient background on the causes of oxidative stress and cytokines in microglia cells and how to mitigate these effects (page 4)- Molecular Neurobiology 57, 1085 (2020).
  2. Since this is a review type of manuscript, figures and tables are more straightforward.

Author Response

Dear reviewer,

The authors sought to provide a detailed view of Ab in AD. I felt that a lot of effort was contributed to achieve this goal, and this manuscript is clear logic and well-written. However, it is mandatory to explain/correct the manuscript in some points.

1) Several important papers should be included in the introduction to provide sufficient background on the causes of oxidative stress and cytokines in microglia cells and how to mitigate these effects (page 4)- Molecular Neurobiology 57, 1085 (2020).

We added the article in the page 4 (see 4 page, ref. [73] in the file "biology-1188761).

2) Since this is a review type of manuscript, figures and tables are more straightforward.

We are appreciated your comment but we not truly understand. We wrote more about description for second picture on 9 page.

We hope that we took into account all your comments.

Also, you can find all our revision in a file "biology-1188761" using the "Track Changes" function in Microsoft Word and see English Editing of our manuscript in a file "After Eng p". The text with all correction, you can see in a file "Final version".

Thus, in this letter you can find 3 documents: one of them is our work with reviewer comments, second is with English Editing, and the third is the final version of the manuscript.

Reviewer 2 Report

I was very please to review the manuscript by Shumeyko et al., Amyloids: the history of toxicity and functionality. Overall the manuscript presents a nice overview of amyloid in its role in Alzheimer's disease. I think this topic is important and timely. The authors do a nice job covering this topic and generally speaking it was quite comprehensive. With that said, I have a few suggestions to improve the manuscript.

First, given that the focus of the topic is amyloid, a little more discussion into the non-pathological role of amyloid would be appropriate. Next, the English grammar in the manuscript needs some improvement. The first 1/4 of the manuscript seems to be the biggest issue, and after that it is generally good. For example, on page two, the sentences "Not only amyloids are found in specific deposits in tissues with amyloidosis. Also, glycosaminoglycans, apolipoprotein E, and serum amyloid P components were found in the deposits." This sentence would be better written as one sentence; for example, "Not only are amyloids found in specific deposits in tissues with amyloidosis, proteins such as glycosaminoglycans, apolipoprotein E, and serum amyloid P components are also often present" There are several instances like this were they English grammar is a bit award and difficult to read. Fixing this would take the paper from good to excellent in my opinion. Lastly, there are a number of typos that need to be fixed. For instance, in the sentence, "To date, amyloids are defined as aggregates of a misfolded peptide or protein", peptide or protein should be plural. Also, PrP needs to be defined at its first use, there is also some places where "in vivo" is italicized and some please it is not. The should be consistent. Lastly, there should be a space between 37 and the degree. 37 °C , not 37°C and one instance where Ca should be Ca2+. This is not a complete proofing but just to show examples. Given how good this paper could be, it is worth the effort to make it perfect. 

Other than that the review is very good and the authors did a nice job.

Author Response

Dear reviewer,

1) I was very pleased to review the manuscript by Shumeyko et al., Amyloids: the history of toxicity and functionality. Overall, the manuscript presents a nice overview of amyloid in its role in Alzheimer's disease. I think this topic is important and timely. The authors do a nice job covering this topic and generally speaking it was quite comprehensive. With that said, I have a few suggestions to improve the manuscript.

We thank for the comment and we tried to improve of our manuscript.

1) First, given that the focus of the topic is amyloid, a little more discussion into the non-pathological role of amyloid would be appropriate.

Thank you for the comment. We fixed it and wrote more about good effects of amyloids. You can find all our revision in a file "biology-1188761" using the "Track Changes" function in Microsoft Word. In particular you can find more discuss about good role of amyloids on page 9, paragraph 1, 2 and on page 16 paragraph 1,2,3.

2) Next, the English grammar in the manuscript needs some improvement. The first 1/4 of the manuscript seems to be the biggest issue, and after that it is generally good. For example, on page two, the sentences "Not only amyloids are found in specific deposits in tissues with amyloidosis. Also, glycosaminoglycans, apolipoprotein E, and serum amyloid P components were found in the deposits." This sentence would be better written as one sentence; for example, "Not only are amyloids found in specific deposits in tissues with amyloidosis, proteins such as glycosaminoglycans, apolipoprotein E, and serum amyloid P components are also often present" There are several instances like this were they English grammar is a bit award and difficult to read. Fixing this would take the paper from good to excellent in my opinion.

Thank you for the comment. We used the English Editing service and you can see the proving in the file "After Eng p" and see the final version with all correction in a file "Final version".

3) Lastly, there are a number of typos that need to be fixed. For instance, in the sentence, "To date, amyloids are defined as aggregates of a misfolded peptide or protein", peptide or protein should be plural. Also, PrP needs to be defined at its first use, there is also some places where "in vivo" is italicized and some please it is not. The should be consistent. Lastly, there should be a space between 37 and the degree. 37 °C , not 37°C and one instance where Ca should be Ca2+. This is not a complete proofing but just to show examples. Given how good this paper could be, it is worth the effort to make it perfect.

Thank you for the comment, we checked your examples and have did many corrections of the text. You can find all in a file "biology-1188761" using the "Track Changes" function in Microsoft Word.

Other than that the review is very good and the authors did a nice job.

Thank you very much.

Best wishes,

The authors

Reviewer 3 Report

The manuscript deals with the amyloid effects on neurodegeneration, focusing on Alzheimer’s disease and the functional use of “amyloids” against bacteria as well as in the treatment of cancer”, as reported by the authors in the abstract.

My main concern is the gap between the title:  “Amyloids: the History of Toxicity and Functionality” and the content of manuscript. Among the different sections of the manuscript, the first three (including Introduction) are mainly dedicated to Abeta and Alzheimer’s disease (AD). Many and recent reviews have been published on the role of Abeta in AD, while sections in the manuscript result unfit to account of the debate about the limits of amyloid cascade hypothesis.  One section alone is focused on functional amyloids and reports in a table one long list of functional amyloids in different biological systems. No relation between bacterial amyloids and neurodegeneration is discussed, except one citation, the N.169 quoted reference.

The above-mentioned points of weakness of the manuscript justify the reason for which it cannot be recommended for publication. The authors can find a partly list of papers that could help them to overcome the limits of this submitted version.

Partial Prion Cross-Seeding between Fungal and ammalian Amyloid Signaling Motifs

Thierry Bardin, Asen Daskalov, Sophie Barrouilhet, Alexandra Granger-Farbos, Bénédicte Salin, Corinne Blancard, Brice Kauffmann, Sven J. Saupe, Virginie Coustou

mBio. 2021 Jan-Feb; 12(1): e02782-20. Published online 2021 Feb 9. doi: 10.1128/mBio.02782-20

The Immunopathogenesis of Alzheimer’s Disease Is Related to the Composition of Gut Microbiota

Friedrich Leblhuber, Daniela Ehrlich, Kostja Steiner, Simon Geisler, Dietmar Fuchs, Lukas Lanser, Katharina Kurz

Nutrients. 2021 Feb; 13(2): 361. Published online 2021 Jan 25. doi: 10.3390/nu13020361

The expanding scope of amyloid signalling

Asen Daskalov, Sven J. Saupe

Prion. 2021; 15(1): 21–28. Published online 2021 Feb 12. doi: 10.1080/19336896.2021.1874791

Biological Functions of Prokaryotic Amyloids in Interspecies Interactions: Facts and Assumptions

Anastasiia O. Kosolapova, Kirill S. Antonets, Mikhail V. Belousov, Anton A. Nizhnikov

Int J Mol Sci. 2020 Oct; 21(19): 7240. Published online 2020 Sep 30. doi: 10.3390/ijms21197240

Accumulation of storage proteins in plant seeds is mediated by amyloid formation

Kirill S. Antonets, Mikhail V. Belousov, Anna I. Sulatskaya, Maria E. Belousova, Anastasiia O. Kosolapova, Maksim I. Sulatsky, Elena A. Andreeva, Pavel A. Zykin, Yury V. Malovichko, Oksana Y. Shtark, Anna N. Lykholay, Kirill V. Volkov, Irina M. Kuznetsova, Konstantin K. Turoverov, Elena Y. Kochetkova, Alexander G. Bobylev, Konstantin S. Usachev, Oleg. N. Demidov, Igor A. Tikhonovich, Anton A. Nizhnikov

PLoS Biol. 2020 Jul; 18(7): e3000564. Published online 2020 Jul 23. doi: 10.1371/journal.pbio.3000564

Functional Amyloids Are the Rule Rather Than the Exception in Cellular Biology

Anthony Balistreri, Emily Goetzler, Matthew Chapman

Microorganisms. 2020 Dec; 8(12): 1951. Published online 2020 Dec 9. doi: 10.3390/microorganisms8121951

Multifunctional Amyloids in the Biology of Gram-Positive Bacteria

Ana Álvarez-Mena, Jesús Cámara-Almirón, Antonio de Vicente, Diego Romero

Microorganisms. 2020 Dec; 8(12): 2020. Published online 2020 Dec 17. doi: 10.3390/microorganisms8122020

Bacterial amyloids: the link between bacterial infections and autoimmunity

Lauren Nicastro, Çagla Tükel

Trends Microbiol. Author manuscript; available in PMC 2020 Nov 1.

Published in final edited form as: Trends Microbiol. 2019 Nov; 27(11): 954–963. Published online 2019 Aug 15. doi: 10.1016/j.tim.2019.07.002

Amyloid β oligomers inhibit growth of human cancer cells

Bozena Pavliukeviciene, Aiste Zentelyte, Marija Jankunec, Giedre Valiuliene, Martynas Talaikis, Ruta Navakauskiene, Gediminas Niaura, Gintaras Valincius

PLoS One. 2019; 14(9): e0221563. Published online 2019 Sep 11. doi: 10.1371/journal.pone.0221563

The Role of Functional Amyloids in Bacterial Virulence

Nani Van Gerven, Sander E. Van der Verren, Dirk M. Reiter, Han Remaut

J Mol Biol. 2018 Oct 12; 430(20): 3657–3684. doi: 10.1016/j.jmb.2018.07.010

Why Are Functional Amyloids Non-Toxic in Humans?

Matthew P. Jackson, Eric W. Hewitt

Biomolecules. 2017 Dec; 7(4): 71. Published online 2017 Sep 22. doi: 10.3390/biom7040071

Functional Amyloids and their Possible Influence on Alzheimer Disease

Angus Lau, Matthew Bourkas, Yang Qing Qin Lu, Lauren Anne Ostrowski, Danielle Weber-Adrian, Carlyn Figueiredo, Hamza Arshad, Seyedeh Zahra Shams Shoaei, Christopher Daniel Morrone, Stuart Matan-Lithwick, Karan Joshua Abraham, Hansen Wang, Gerold Schmitt-Ulms

Discoveries (Craiova) 2017 Oct-Dec; 5(4): e79. Published online 2017 Oct 16. doi: 10.15190/d.2017.9

Functional amyloids in Streptococcus mutans, their use as targets of biofilm inhibition and initial characterization of SMU_63c

Richard N Besingi, Iwona B Wenderska, Dilani B Senadheera, Dennis G Cvitkovitch, Joanna R Long, Zezhang T Wen, L. Jeannine Brady

Microbiology (Reading) 2017 Apr; 163(4): 488–501. Published online 2017 Apr 26. doi: 10.1099/mic.0.000443

The role of microbial amyloid in neurodegeneration

Robert P. Friedland, Matthew R. Chapman

PLoS Pathog. 2017 Dec; 13(12): e1006654. Published online 2017 Dec 21. doi: 10.1371/journal.ppat.1006654

Prions, amyloids, and RNA: Pieces of a puzzle

Anton A. Nizhnikov, Kirill S. Antonets, Stanislav A. Bondarev, Sergey G. Inge-Vechtomov, Irina L. Derkatch

Prion. 2016 May-Jun; 10(3): 182–206. Published online 2016 Jun 1. doi: 10.1080/19336896.2016.1181253

Antimicrobial Properties of Amyloid Peptides

Bruce L. Kagan, Hyunbum Jang, Ricardo Capone, Fernando Teran Arce, Srinivasan Ramachandran, Ratnesh Lal, Ruth Nussinov

Mol Pharm. Author manuscript; available in PMC 2013 Apr 2.

Published in final edited form as: Mol Pharm. 2012 Apr 2; 9(4): 708–717. Published online 2011 Nov 29. doi: 10.1021/mp200419b

Author Response

Dear reviewer,

1) The manuscript deals with the amyloid effects on neurodegeneration, focusing on Alzheimer’s disease and the functional use of “amyloids” against bacteria as well as in the treatment of cancer”, as reported by the authors in the abstract.

Thank you for the comment. We change the Abstract for more understanding about the text of the manuscript. You can see a new version on one page of a file "biology-1188761" using the "Track Changes" function in Microsoft Word.

My main concern is the gap between the title: “Amyloids: the History of Toxicity and Functionality” and the content of manuscript. Among the different sections of the manuscript, the first three (including Introduction) are mainly dedicated to Abeta and Alzheimer’s disease (AD). Many and recent reviews have been published on the role of Abeta in AD, while sections in the manuscript result unfit to account of the debate about the limits of amyloid cascade hypothesis.  One section alone is focused on functional amyloids and reports in a table one long list of functional amyloids in different biological systems. No relation between bacterial amyloids and neurodegeneration is discussed, except one citation, the N.169 quoted reference.

Thank you for the comment. We see your concerning. We mainly dedicated the review to Abeta peptide toxicity because "Aβ-amyloid is the most studied representative of amyloids. Therefore, in this review, special attention is paid to Aβ-amyloid" that you can see on page 2, paragraph 4. Of course, we did not want to say that there is relation between bacterial amyloids and neurodegeneration. Due to your comment, we wrote more about functional amyloids in the text. You can find all our revision in a file "biology-1188761" using the "Track Changes" function in Microsoft Word. In particular you can find more discuss about good role of amyloids on page 9, paragraph 1, 2 and on page 16 paragraph 1,2,3.

2) The above-mentioned points of weakness of the manuscript justify the reason for which it cannot be recommended for publication. The authors can find a partly list of papers that could help them to overcome the limits of this submitted version.

Thank you for the advice. We can notice that we used more that 200 references for the manuscript. We saw your reference list and added 14. For instance, see ref. 183, 186, 187, 188, 189, 190, 241, 244, 245, 246. The final number is 246 references in the article.

Thank you for your work,

Best wishes,

Authors.

Round 2

Reviewer 3 Report

The revised version of the manuscript can be recommended for publication

This manuscript is a resubmission of an earlier submission. The following is a list of the peer review reports and author responses from that submission.

Round 1

Reviewer 1 Report

The authors did a good homework and improved the manuscript according to reviewer's recommendations.

Reviewer 2 Report

This manuscript has been sent for consideration as a review. Amyloid fibril formation has been reported by Chris Dobson as a general property of almost every polypeptide chains. As described by the authors some amyloid are directly involved in dramatic diseases for which there is no therapeutic issue yet. This review tends to describe the history of amyloid toxicity and functionality. In my opinion, this was very hazardous to mix in the same work functional and non-functional amyloid. The authors focus on Abeta peptide and also on different functional amyloid as described in Table 1. It results in a confusing review, where it is quite hard to follow the main line. Moreover, as the authors have decided to describe the history, many described studies referred to early work on the toxicity of fibrils. Investigation in this field has moved during the last ten years towards oligomerization as the main toxicity reason.

For all these reasons, I would not recommend publication of this review in IJMS.